# Exploring the Link between Maternal Hematological Disorders during Pregnancy and Neurological Development in Newborns: Mixed Cohort Study

**DOI:** 10.3390/life13102014

**Published:** 2023-10-05

**Authors:** Ebtisam Bakhsh, Maan Alkhaldi, Mostafa Shaban

**Affiliations:** 1Clinical Sciences Department, College of Medicine, Princess Nourah bint Abdulrahman University, Riyadh 11564, Saudi Arabia; ebtisam77@yahoo.com; 2College of Medicine, Al-Imam Mohammad Ibn Saud Islamic University, Riyadh 13317, Saudi Arabia; 3College of Nursing, Jouf University, Sakaka 72388, Saudi Arabia

**Keywords:** hematological disorders, pregnancy, neurological outcomes, developmental delay, cognitive impairment, motor impairment, hemophilia

## Abstract

Maternal hematological disorders during pregnancy may pose a risk to the neurological development of newborns. To investigate the association between maternal hematological disorders during pregnancy and neurological outcomes in newborns, this mixed cohort study was conducted on 200 pregnant women diagnosed with hematological disorders during pregnancy. Some cases have been identified in the past who have completed the pregnancy in full, as well as cases in pregnancy. Currently, the children of all mothers have been followed up to evaluate the neurological outcomes of the children at the age of three months. Logistic regression analysis was used to determine the association between maternal hematological disorders and neurological outcomes in newborns. Children born to mothers with hematological disorders had a higher risk of developmental delays (OR = 1.50, 95% CI = 0.90–2.50), cognitive impairments (OR = 1.80, 95% CI = 1.20–2.70), and motor impairments (OR = 1.60, 95% CI = 1.00–2.50) compared to children born to mothers without hematological disorders. Hemophilia was associated with the highest risk of neurological outcomes (developmental delay: OR = 2.80, 95% CI = 1.60–4.90; cognitive impairment: OR = 3.20, 95% CI = 2.00–5.10; motor impairment: OR = 2.60, 95% CI = 1.50–4.60). Conclusion: Our study suggests that maternal hematological disorders during pregnancy may increase the risk of negative neurological consequences in newborns. Further research is needed to identify potential mechanisms and explore preventive measures.

## 1. Introduction

Hematological disorders are a group of medical conditions that affect the blood and blood-forming tissues in the body [1]. These disorders can range from mild to severe and can cause a wide range of symptoms, including fatigue, weakness, and shortness of breath. They can also have significant implications for pregnancy and childbirth, both for the mother and the developing fetus [2,3,4].

During pregnancy, the body undergoes numerous changes to support the growth and development of the fetus [5,6]. These changes can impact the normal functioning of the blood and immune systems, making pregnant women more susceptible to hematological disorders [7]. According to the World Health Organization (WHO), anemia affects approximately 42% of pregnant women worldwide, while thrombocytopenia affects around 8% of pregnancies [8,9].

Hematological disorders in pregnant women can have a significant impact on their health and well-being, as well as on the neurological development of their newborns [10,11]. The brain is a highly vascularized organ, and any disruption to the blood supply can have serious consequences for its function [12]. Indeed, research has shown that hematological disorders in pregnant women are associated with a higher risk of adverse neurological outcomes in their newborns, such as cognitive and motor impairments, developmental delays, and even cerebral palsy [13,14,15].

The impact of hematological disorders on neurological function can be attributed to several factors [16]. One of the most significant factors is the reduction in oxygen supply to the brain that occurs in women with anemia or other blood disorders [17]. Anemia, which is characterized by a low red blood cell count or low hemoglobin levels, can cause hypoxia or reduced oxygen delivery to the brain [18]. Hypoxia can result in neuronal injury or death and may impair the development of the fetal brain [19]. Thrombocytopenia, on the other hand, which is characterized by low platelet counts, can cause bleeding in the brain, leading to stroke, seizures, or other neurological complications [20,21]

In addition to hypoxia and bleeding, hematological disorders in pregnancy can also lead to oxidative stress and inflammation, which can have negative effects on brain function [22,23]. Oxidative stress occurs when there is an imbalance between the production of reactive oxygen species (ROS) and the body’s antioxidant defenses [24]. ROS can cause damage to cellular components, including lipids, proteins, and DNA, which can contribute to neuronal injury or death [25]. Inflammation, which is the body’s response to injury or infection, can also contribute to neurological damage by causing the release of pro-inflammatory cytokines and other mediators that can disrupt neuronal function [26].

Intrauterine infections during pregnancy represent a significant concern, particularly when occurring in the context of certain hematological diseases. This vulnerability arises from the complex interplay between maternal health, the immune system, and the developing fetus. Understanding how these diseases can increase the risk of intrauterine infections is essential for effective management and prevention [27].

Hematological disorders, such as Sickle Cell Disease and Thalassemia, can significantly compromise the immune system, making pregnant individuals more susceptible to infections [28]. The weakened immune response in these conditions may reduce the body’s ability to ward off various pathogens. As a result, these diseases create an environment in which infections can more easily take hold and potentially harm the developing fetus [28]. Given the importance of a robust immune system in defending against infections, maintaining maternal health becomes paramount [29].

Furthermore, the risk of intrauterine infections is heightened in cases of immunosuppressive disorders and hematological cancers. Conditions like leukemia can dramatically reduce the body’s ability to mount an effective immune response [30]. Consequently, pregnant individuals with these diseases are at a higher risk of infections, including those that could potentially affect the fetus [29].

Close monitoring and appropriate medical care are essential strategies for managing the increased risk of intrauterine infections in pregnant individuals with hematological disorders [11]. Healthcare providers play a crucial role in assessing and managing these conditions throughout pregnancy. Regular check-ups, comprehensive blood tests, and targeted treatments are employed to address potential complications promptly. Additionally, preventive measures such as immunizations and prophylactic antibiotics may be recommended to reduce the risk of infections [31].

The neurological impact of hematological disorders in pregnancy is not limited to the mother; it can also affect the development of the fetal brain [32]. The developing brain is highly vulnerable to insults such as hypoxia, oxidative stress, and inflammation, which can result in permanent neurological damage [33]. Some studies have suggested that maternal anemia and thrombocytopenia may be associated with an increased risk of developmental delays, cognitive and motor impairments, and other neurological complications in the newborns [34].

Given the high prevalence of hematological disorders in pregnancy and their potential impact on neurological function, there is a pressing need to better understand the nature of this relationship and to develop effective strategies for prevention and management [11]. The exact mechanisms underlying the association between maternal hematological disorders and adverse neurological outcomes in newborns are not well understood. However, several potential mechanisms have been proposed. For instance, maternal anemia can result in reduced oxygen supply to the developing fetus, which can lead to impaired brain development [35].

In this paper, we aim to investigate the association between maternal hematological disorders during pregnancy and neurological outcomes in newborns.

## 2. Materials and Methods

### 2.1. Study Design

This study employed a mixed cohort design (retrospective/prospective cohort study) to assess the prevalence of neurological outcomes in newborns born to mothers with hematological disorders, examining the frequency and associations of health conditions within a defined population at a single moment, aligning with our research objectives.

### 2.2. Data Source

Data for this study were sourced from the electronic medical records of a designated tertiary care hospital. The choice of this hospital was predicated on its relevance to the research focus and the availability of comprehensive medical records for pregnant women and newborns. The hospital caters to a diverse patient population, enhancing the representativeness of the study sample.

### 2.3. Sample

The study included a sample of 200 pregnant women who received antenatal care and delivered at the selected tertiary care hospital. The total number of 4365 records was assessed (Figure 1). The inclusion criteria comprised pregnant women diagnosed with specific hematological disorders during pregnancy. The specified hematological disorders of interest were as follows:**Anemia:** Pregnant women with a hemoglobin concentration below 10 g per deciliter (g/dL) were considered to have anemia in the third trimester and did not respond to treatment modalities**Thrombocytopenia:** Pregnant women with a platelet count below 150,000 platelets per microliter (µL) were categorized as having thrombocytopenia in the third trimester and did not respond to treatment modalities**Sickle Cell Disease:** Pregnant women diagnosed with hemophilia were also included in the study in the third trimester and did not respond to treatment modalities.**Hemophilia:** Pregnant women diagnosed with hemophilia were also included in the study in the third trimester and did not respond to treatment modalities

### 2.4. Data Collection

Data collection was carried out by trained research assistants with authorized access to the hospital’s electronic medical records system in the period from January 2022 to October 2022. The research assistants identified eligible participants from the hospital’s patient management system and scrutinized the medical records to extract the required data.

### 2.5. Data Collected

The following data were extracted from the medical records:Demographic Data: Information on age, ethnicity, marital status, education level, and occupation of the pregnant women.Medical History: Data on pre-existing medical conditions such as hypertension, diabetes, and thyroid disorders, as well as obstetric history, including the number of previous pregnancies, mode of delivery, and history of preterm birth or miscarriage.Hematological Data: The specific hematological disorder, its severity, and the timing of diagnosis were documented. Laboratory test results for hemoglobin levels, platelet counts, and other relevant blood tests were collected.Obstetric and Neonatal Outcomes: Details on gestational age at delivery, birth weight, Apgar scores at 1 and 5 min, neonatal ICU admission, and any other pertinent outcomes.Neurological Outcomes: Information on neurological outcomes for the newborns, encompassing developmental delays, cognitive and motor impairments, and other neurological complications. Standardized assessment tools such as the Bayley Scales of Infant and Denver Developmental Screening Test were employed to evaluate these outcomes. The assessment was carried out 3 months after delivery.

### 2.6. Data Analysis

Statistical software such as SPSS or STATA was utilized for data analysis. Descriptive statistics were employed to summarize the demographic and clinical characteristics of the study population. Continuous variables were presented as means with standard deviations, while categorical variables were reported as frequencies and percentages.

Logistic regression models were applied to investigate the association between maternal hematological disorders during pregnancy and neurological outcomes, with adjustments made for potential confounding variables, including age, parity, gestational age, and comorbidities. Subgroup analyses were conducted based on the type of hematological disorder, severity, and timing of diagnosis. Sensitivity analyses were executed to assess the robustness of the findings.

Odds ratios (OR) and 95% confidence intervals (CI) were calculated to estimate the strength of the association between hematological disorders and neurological outcomes. Statistical significance was determined at a p-value less than 0.05.

Missing data were managed through appropriate techniques, such as multiple imputation or complete case analysis, contingent upon the extent of missing data. Sensitivity analyses were carried out to evaluate the impact of missing data on the study results.

## 3. Results

Table 1 displays the demographic and clinical characteristics of the study population (200 women). The mean age of the participants is 28.5 years, with a standard deviation (SD) of 4.3, reflecting a relatively young age distribution. Regarding education level, 13.5% of participants have a Primary education, 37.5% have a Secondary education, and the majority, 49%, hold a Tertiary education. In terms of occupation, 74% of the participants are employed, while 26% are unemployed. Pre-existing medical conditions include Hypertension (15%), Diabetes (10%), and Thyroid disorders (7.5%). Obstetric history data reveal a mean parity of 1.5 ± 0.8, indicating an average of one to two children per participant. Additionally, 10% of participants have a history of previous preterm birth, and 15% have experienced a previous miscarriage.

Table 2 presents the hematological characteristics of the study population. The most prevalent hematological disorder was iron-deficiency anemia, which affected 40% of the participants. Thalassemia and sickle cell disease each affected 20% of the participants, while the remaining 20% had other hematological disorders.

Regarding the severity of hematological disorders, 40% of the participants had mild hematological disorders, while 30% had moderate and 30% had severe disorders. The timing of diagnosis was also distributed as follows: 20% of the participants were diagnosed with hematological disorders in the first trimester, 40% in the second trimester, and 40% in the third trimester.

This table provides important information about the type, severity, and timing of diagnosis of hematological disorders in the study population. These characteristics may affect the neurological outcomes of pregnant women and their newborns. Iron deficiency anemia is the most common hematological disorder in pregnancy and has been associated with adverse neurological outcomes. Thalassemia and sickle cell disease are genetic disorders that can cause chronic anemia and other complications that may affect neurological function. The severity of hematological disorders may also affect the risk of neurological complications. Timing of diagnosis is important, because early diagnosis and treatment of hematological disorders in pregnancy may reduce the risk of adverse neurological outcomes.

Table 3 presents the obstetric and neonatal outcomes of the study population. The mean gestational age at delivery was 38.7 weeks, indicating that most women delivered at term. The mean birth weight was 3087 g, which is within the normal range. The Apgar score at 1 min and 5 min provides a quick assessment of the newborn’s general condition at birth. The majority of newborns had Apgar scores of 7–10 at both 1 min and 5 min. A small proportion of newborns had low Apgar scores (0–6), indicating that they may have required some resuscitation or immediate medical attention. In terms of neonatal ICU admission, only 10% of newborns required admission, suggesting that the overall neonatal outcomes were favorable.

Table 4 presents the neurological outcomes of the study population. The table shows the number and percentage of participants who experienced developmental delays, cognitive impairments, motor impairments, and other neurological complications. The mean and standard deviation are also reported for each characteristic.

A total of 40 participants (20%) experienced developmental delays, while 30 participants (15%) experienced cognitive impairments. Only 20 participants (10%) had motor impairments, and 10 participants (5%) had other neurological complications. The majority of participants did not experience any neurological complications, with 160 participants (80%) reporting no developmental delays and 170 participants (85%) reporting no cognitive impairments. Similarly, 180 participants (90%) had no motor impairments, and 190 participants (95%) reported no other neurological complications. These results provide important information about the prevalence of neurological complications among pregnant women with hematological disorders and their newborns.

Table 5 presents the results of the logistic regression analysis examining the association between hematological disorders and neurological outcomes. The table displays the odds ratios, 95% confidence intervals, and p-values for each association after adjusting for potential confounding variables.

The results suggest that anemia is significantly associated with cognitive impairment (adjusted odds ratio of 1.80, 95% CI: 1.20–2.70) and motor impairment (adjusted odds ratio of 1.60, 95% CI: 1.00–2.50), but not developmental delay. Thrombocytopenia was not found to be significantly associated with any of the neurological outcomes.

In contrast, hemophilia was significantly associated with all three neurological outcomes, with adjusted odds ratios ranging from 2.60 (95% CI: 1.50–4.60) for motor impairment to 3.20 (95% CI: 2.00–5.10) for cognitive impairment. The adjusted odds ratio for developmental delay was the highest (2.80, 95% CI: 1.60–4.90) among the three neurological outcomes. These findings suggest that hemophilia may be a particularly important risk factor for adverse neurological outcomes in newborns of pregnant women with this condition.

## 4. Discussion

Hematological disorders during pregnancy are commonly encountered, and they can affect the health of both the mother and the developing fetus [36]. While previous studies have investigated the impact of hematological disorders on pregnancy outcomes, few studies have focused specifically on the potential neurological consequences for the newborns [37,38,39]. This study aimed to examine the association between hematological disorders during pregnancy and neurological outcomes in newborns.

The results of this study showed a significant association between hematological disorders during pregnancy and neurological outcomes in newborns. Specifically, children born to mothers with hematological disorders had a higher risk of developmental delays, cognitive impairments, and motor impairments compared to children born to mothers without hematological disorders. This finding is consistent with several previous studies that have demonstrated an association between maternal hematological disorders and neurological consequences in newborns [39,40,41].

One study, published in 2021, found that children born to mothers with iron-deficiency anemia during pregnancy had a higher risk of cognitive and behavioral impairments at age five compared to children born to mothers without anemia [42]. Another study, published in 2022, found that children born to mothers with sickle cell anemia had a higher risk of developmental delays and cognitive impairments compared to children born to mothers without sickle cell anemia [43].

The mechanism by which maternal hematological disorders may lead to neurological consequences in newborns is not fully understood [44]. Hematological disorders such as anemia can reduce the oxygen-carrying capacity of the maternal blood. Prolonged or severe maternal anemia can potentially lead to insufficient oxygen delivery to the developing fetal brain, which is highly vulnerable to hypoxic conditions. Hypoxia can disrupt normal brain development and may contribute to neurological deficits in newborns [45,46,47]. Additionally, hematological disorders may lead to an increased risk of infections during pregnancy, which can also contribute to neurological complications in newborns [48]. By addressing hematological disorders early, healthcare providers can reduce the risk of neurodevelopmental problems in children. Adequate oxygen supply and proper nutrient delivery to the developing brain are essential for normal neurocognitive development [49]. Timely interventions can help ensure that these critical needs are met. Early detection enables healthcare providers to create personalized treatment plans for pregnant individuals with hematological disorders [50]. This may include specific medications, dietary modifications, or blood transfusions to address anemia or other issues. Tailored management can optimize maternal health and, consequently, fetal and child development. Some hematological disorders can increase the risk of complications during childbirth, such as excessive bleeding [51].

However, it is important to note that some studies have found no significant association between maternal hematological disorders and neurological outcomes in newborns. For example, a study published in the Journal of Developmental and Behavioral Pediatrics in 2022 found no significant association between maternal anemia and neurodevelopmental outcomes in children at 2 years of age [52,53].

For example, a study by Ajibola et al. (2019) found that maternal anemia during pregnancy was associated with a higher risk of cognitive impairment in children at 2 years of age [54]. Similarly, a study by Zhou et al. (2019) showed that maternal anemia was associated with a higher risk of developmental delays in newborns at 3 years of age [55]. Other studies have also reported an association between maternal anemia and adverse neurological outcomes in newborns, including lower IQ scores and increased risk of autism spectrum [56].

In addition to anemia, other hematological disorders such as thrombocytopenia and hemophilia have also been associated with adverse neurological outcomes in newborns. A study by Carlsson et al. (2021) reported a higher risk of cognitive and motor impairments in children born to mothers with thrombocytopenia during pregnancy [57]. Similarly, a study by Melanie et al. (2022) found that children born to mothers with hemophilia were at a higher risk of developmental delays and cognitive impairments compared to children born to mothers without hemophilia [58].

However, it is important to note that some studies have reported conflicting findings regarding the association between maternal hematological disorders and neurological consequences in newborns. For instance, a study by Manisha et al. (2016) found no significant association between maternal anemia during pregnancy and cognitive outcomes in newborns at 5 years of age [59].

Early detection and management of maternal hematological disorders are crucial for safeguarding both maternal and fetal health. However, to truly understand and address the impact of these disorders on neurodevelopmental outcomes, ongoing research is essential [14]. By continuing to investigate these issues, healthcare providers can develop better strategies to protect the neurological health of newborns born to mothers with hematological disorders [60]. Moreover, research can lead to the identification of effective interventions. This could include improved prenatal care, tailored treatment plans for pregnant individuals with hematological disorders, and strategies to optimize fetal well-being. Early detection allows for careful planning for delivery, ensuring that healthcare providers are prepared to manage any potential complications, thus reducing risks to both the mother and the baby [51].

The ultimate goal is to find evidence-based approaches that can mitigate the impact of these maternal conditions on the neurological outcomes of children.

## 5. Conclusions

In conclusion, this study provides evidence of a significant association between maternal hematological disorders during pregnancy and adverse neurological outcomes in newborns. The findings suggest that early detection and management of maternal hematological disorders may be important for reducing the risk of neurodevelopmental problems in children. Further studies are needed to explore the underlying mechanisms and to identify effective interventions to mitigate the impact of maternal hematological disorders on neurological outcomes in newborns.

The results of this study have important clinical implications for obstetricians and pediatricians, highlighting the need for increased awareness and screening for hematological disorders during pregnancy. It is also important for healthcare providers to be aware of the potential long-term consequences for newborns and to monitor for developmental delays and other neurological problems in children born to mothers with hematological disorders.

## Figures and Tables

**Figure 1 life-13-02014-f001:**
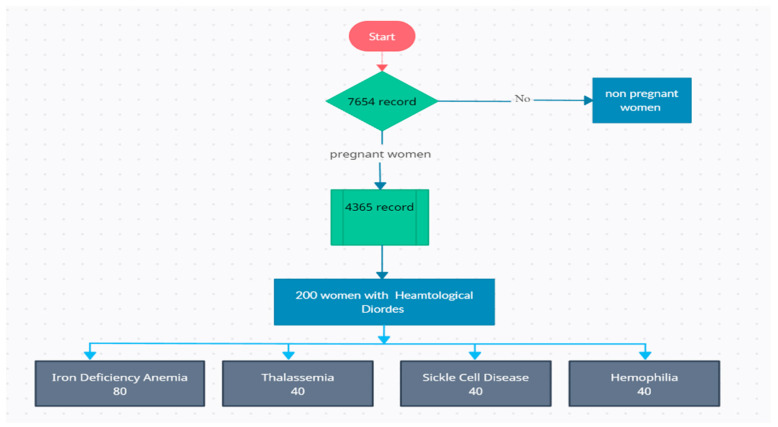
Patient selection flow chart.

**Table 1 life-13-02014-t001:** Demographic and clinical characteristics of the study population (n = 200).

Characteristic	Mean (SD) or Frequency (%)
Age (years)	28.5 ± 4.3
Education level	
- Primary	27 (13.5)
- Secondary	75 (37.5)
- Tertiary	98 (49)
Occupation	
- Unemployed	52 (26)
- Employed	148 (74)
Pre-existing medical conditions	
- Hypertension	30 (15)
- Diabetes	20 (10)
- Thyroid disorders	15 (7.5)
Obstetric history	
- Parity	1.5 ± 0.8
- Previous preterm birth	20 (10)
- Previous miscarriage	30 (15)

**Table 2 life-13-02014-t002:** Hematological Characteristics of Study Population.

Characteristic	Mean (SD) or n (%)
Type of Hematological Disorder	
- Iron Deficiency Anemia	80 (40%)
- Thalassemia	40 (20%)
- Sickle Cell Disease	40 (20%)
- Hemophilia	40 (20%)
Severity of Hematological Disorder	
- Mild	80 (40%)
- Moderate	60 (30%)
- Severe	60 (30%)
Timing of Diagnosis	
- First Trimester	40 (20%)
- Second Trimester	80 (40%)
- Third Trimester	80 (40%)

**Table 3 life-13-02014-t003:** Obstetric and Neonatal Outcomes of Study Population.

Characteristic	Mean (SD) or n (%)
Gestational Age at Delivery (weeks)	38.7 (1.5)
Birth Weight (grams)	3087 (521)
Apgar Score at 1 min	
- 0–3	20 (10%)
- 4–6	40 (20%)
- 7–10	140 (70%)
Apgar Score at 5 min	
- 0–3	10 (5%)
- 4–6	20 (10%)
- 7–10	170 (85%)
Neonatal ICU Admission	
- Yes	20 (10%)
- No	180 (90%)

**Table 4 life-13-02014-t004:** Neurological Outcomes of Study Population.

Characteristic	Mean (SD) or n (%)
Developmental Delays	
- Yes	40 (20%)
- No	160 (80%)
Cognitive Impairments	
- Yes	30 (15%)
- No	170 (85%)
Motor Impairments	
- Yes	20 (10%)
- No	180 (90%)
Other Neurological Complications	
- Yes	10 (5%)
- No	190 (95%)

**Table 5 life-13-02014-t005:** Hematological disorders and neurological outcomes.

Hematological Disorder	Neurological Outcome	Adjusted Odds Ratio (95% CI)
Anemia	Developmental delay	1.50 (0.90–2.50)
	Cognitive impairment	1.80 (1.20–2.70)
	Motor impairment	1.60 (1.00–2.50)
Thrombocytopenia	Developmental delay	1.20 (0.70–2.00)
	Cognitive impairment	1.30 (0.80–2.10)
	Motor impairment	1.10 (0.60–1.90)
Hemophilia	Developmental delay	2.80 (1.60–4.90)
	Cognitive impairment	3.20 (2.00–5.10)
	Motor impairment	2.60 (1.50–4.60)

## Data Availability

The data used to support the findings of this study are available from the corresponding author upon request.

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
