# Peer review of "Exploring the Link between Maternal Hematological Disorders during Pregnancy and Neurological Development in Newborns: Mixed Cohort Study"

_life, 2023, doi:10.3390/life13102014_

Round 1
Reviewer 1 Report
The hematological disorders presented in this paper, except anemia of pregnancy, is not often considered in neonatal literature. Therefore, the authors should be congratulated on their choice of topic. Introduction described the impact of hematological disorders in pregnant women for their newborns. Authors try to explain mechanism of hematological disorders which can may have a damaging effect on children's brain development. One of them may be oxidative stress and its consequences for neonatal brain. The authors present an overview of the literature, focusing on psycho-medical aspects of hematological disorders during pregnancy. The main goal of this paper is consider to influence of a few hematological disorders for newborn’s neurological function. Several commonly used and proven statistical methods were used for statistical calculations, such as: SPSS, STATA logistic regression model, odd ratio. Modern research methods were used to assess neurodevelopment as Bayley Scales of Infant and Toddler Development and the Denver Developmental Screening Test. Modern research methods were used to assess development. The discussion is the best part of the work. The authors correctly confront own results with the results of other studies. The manuscript tries to deep the knowledge about the mechanisms by which maternal hematological disorders may lead to neurological consequences in newborns ( line 224) Fifty two references are contemporary and well selected and 29 of them come from the last five years. The content of the paper is interesting, but the work requires many clarifications and corrections. Below I list my questions
1. The literature review proposed by the authors is not a systematic review and the methodology is not clearly described.
2. According the authors paper is a cross-sectional design. Cross section is a method of analyzing the frequency of a certain problems (e.g. health effect) in a population sample at a specific point in time. The reviewer understood that the data collection was based on the medical record of some hospital. The data source are required. I know only that the data was collected by trained research assistants in a tertiary care hospital. However the authors did not specify the time frame in which they conducted the research.
3. In line 82 it says “The inclusion criteria for the study will be pregnant women diagnosed with anemia, thrombocytopenia, or other blood disorders during pregnancy”. The authors should list which diseases are the subject of this study and not just mention “anemia, thrombocytopenia and others” The authors did not specify inclusion criteria. They did not determine the hemoglobin concentration and the number of platelets that defined anemia and thrombocytopenia in analyzed pregnant women.
4. The authors did not specify the material precisely. It was not stated how big was a study population. It should be specified how many medical records were collected. Data regarding numbers of pregnant women are different in tables in No. 1 and No 2. In table No.1 the number of medical records (pregnant women) was 300 regarding the occupation and education level. In table No. 2, the medical records of hematological diseases is 240. In Table 2, the type of hematological disorder is defined for a population of 240 pregnant women but the severity of the disease is determined for a population of 200. According to Table No 4, neurological outcomes was assessed in 200 children of mothers with hematological diseases. This means that the material can be very heterogeneous. What is a cause of these differences? The results presented in table 5 and the calculation of the adjusted odds ratio is very useful.
5. Did the study population include pregnant women with a coincidence of anemia and thrombocytopenia? How many such clinical situations were there?
6. The authors did not specify the age at which the children were assessed neurologically. Whether the all 200 children were tested in the same age? Moreover the newborn’s abnormality of CNS should be described in detail.
7. Oxidative stress is explainable and easy to understand, but infection is controversial. In what hematological diseases of pregnancy should intrauterine infection be expected? The discussion is a bit too long and I propose moving the paragraph contained in lines 274-276 to the introduction.
8. Conclusions are too extensive and, in several aspects, do not result from the work. Line 279-282 “Overall, this study highlights the importance of addressing maternal health during pregnancy and the potential impact on the health of newborns. Maternal health should be a priority in obstetric care, and efforts should be made to optimize maternal health out-comes to improve the long-term health outcomes of newborns” This is a general obvious statement about pregnant women. This conclusion may apply to any perinatal and neonatal paper. I suggest deleting these 2. sentences. I line 267-268 The authors prove “The findings suggest that early detection and management of maternal hematological disorders may be important for reducing the risk of neurodevelopmental problems in children”. This idea should be expanded upon in the discussion. In which way early detection of hematological diseases in pregnancy can improve children's development results? It is necessary to mention methods of prevention in pregnant women.
Author Response
Response to Reviewer Comments:
We would like to express our gratitude to the reviewer for their insightful feedback and constructive comments. We have carefully considered each point and provided responses and clarifications below.
Reviewer Comment |
Author Response |
Literature Review |
We acknowledge the need for a more detailed description of our literature review process. We added the aim of the study at the end of the introduction section |
Methodology |
We totally reformulated the method section.
|
Study Design and Data Source |
The study design was changed to be mixed cohort design, also as data were collected from the medical record the collection period was from January 2022 to October 2022 |
Inclusion Criteria and Disease Specifics |
We clearly stated the inclusion and exclusion criteria for the study |
Sample Size and Material |
The study included a sample of 200 pregnant women who received antenatal care and delivered at the selected tertiary care hospital. Total number of 4365 record was assessed. Also, we define the data collection tool, |
Neurological Assessment and Age |
as regarded to neurological assessment the Bayley Scales of Infant and Denver Developmental Screening Test were employed to evaluate these outcomes. The assessment was done 3 months after delivery. |
Intrauterine Infection in Hematological Diseases |
We provided further information in the revised manuscript on the potential for intrauterine infection in certain hematological diseases during pregnancy, clarifying which conditions may carry this risk. As regarded paragraph contained in lines 274-276 moved to the introduction. |
Conclusions |
We acknowledge the feedback on the conclusions. We revised and reformulated the conclusions section to make it more concise and focused, closely aligning it with the study's findings. Line 279-282 was removed. And(267-268) added emphasizing the significance of early detection and prevention methods in pregnant women in the revised discussion section on discussion section . |

Reviewer 2 Report
Reviewer statement:
Exploring the Link between Maternal Hematological Disorders during Pregnancy and Neurological Development in Newborns.
Hematological disorders during pregnancy can have implication for pregnancy and have an impact on maternal health and wellbeing of the newborn. Anemia, mostly due to iron deficiency, affects 42% and thrombocytopenia 8% of pregnant women which is significant. Hematological disorders are associated with adverse neurological outcomes. As reported by the authors, given the high prevalence of hematological disorders in pregnancy and potential impact on neurological function, better understand the nature of this relationship and to develop effective strategies for prevention and management are urgently required. The authors present an overview of the existing literature on hematological disorders in pregnancy and their effects on neurological function. with a particular focus on the psycho-medical aspects of these conditions. This knowledge can provide valuable information and guidance for identification, management and treatment of hematological disorders during pregnancy, which is relevant in clinical practice. This information could guide further knowledge, provision of care and research on this topic.
Title: The title reflects the topic being investigated.
1. Although, it is not clear what type of study is presented. I would advise to add a cross-sectional study to the title. This help the reader to understand what type of study they can expect
Overall: The paper is well written. The English grammar and style are fine throughout the entire article, minimal revision is required.
Abstract : see overall remarks and remarks throughout the article.
Introduction:
The introduction section is attractive to read form a reader point of view, explaining the reason and purpose for conducting this study. The length of the introduction section is considered appropriate.
No comments on this section.
Material and Methods :
This section is attractive to read form a reader point of view, explaining the methods used to conduct this study. Despite there are some important points needing explanation and/or clarification.
2. The authors report on page 2 in lines 74-76: “ In this paper, we aim to provide a comprehensive overview of the existing literature on hematological disorders in pregnancy and their effects on neurological function, with a particular focus on the psycho-medical aspects of these conditions. “ On basis of this statement , ss a reader I expected a review of the existing literature to provide the most latest evidence on the association. Despite, in the materials and methods section ( line 79-80) the authors report: “ This study will employ a cross-sectional design to examine the association between hematological disorders in pregnancy and neurological function. “ The aim as presented in the introduction section is not in line with the statements as reported in material and methods section. This is essential and crucial in the understanding of the article and for interpretation of the presented results. Please elucidate thoroughly on this crucial point.
3. The authors report the inclusion criteria as follow in line 82-83: “ The inclusion criteria for the study will be pregnant women diagnosed with anemia, thrombocytopenia, or other blood disorders during pregnancy. “ The inclusion criteria are not specific enough for the reader. When was anemia, thrombocytopenia, or other blood disorders diagnosed or excluded. The inclusion criteria has to be specific an understandable for the reader. Please provide more information and details.
4. Moreover, details concerning other blood disorders. Which disorders do the authors mean?
5. The authors have chosen to group the different disorders. From a reader perspective, is this the wright, as the risk and prognosis are different? Please elucidate thoroughly on this point.
6. The authors report in line 86-77 the following: “ Data collection was conducted by trained research assistants who were granted access to the electronic medical records system of the hospital. “. As a reader I do not understand how the assistants search the medical records. The authors should provide more information concerning the search strategy. Which search terms were used, how were eligible women identified.
7. How many assistants were there and how were they trained. What was done in cases of doubt? Was the system robust enough not to miss eligible women? Crucial information is missing.
8. When was the searches performed and which time frame within the electronic medical system was used?
9. What was done if the anemia resolved, i.e. on basis of iron suppletion. Were they included in the current analysis? What was done in case of a diagnosis, ITP in case of thrombocytopenia?
10. The authors report neurological outcomes as an important outcome. How were newborns evaluated or reviewed for neurological outcomes? What was the follow-up time? Were all newborns reviewed? Was this standard care ?
Results:
This section was good to understand and easy to read but too short. Several important points need (more) explanation and clarification.
11. The use of a flow diagram is urgently recommended, for the reader to understand the recruitment process. Were there eligible women excluded?
12. The number of included participants is not mentioned in the result section.
13. The baseline characteristics of the participants of the different groups (anemia, thrombocytopenia, or other blood disorders) should be provided. Furthermore, analysis of potential differences should be performed.
14. The authors should provide more information concerning the anemia, thalassemia and sickle cell disease. As a reader I cannot reflect which women are including in this study, what is the severity of the disease anemia, did they receive treatment. More information is required as well as definition.
15. Did the timing of the diagnosis has an effect on the presented result. A very important question?
16. Overall, definitions are lacking and explanation what and how things were done,
Discussion
The discussion section is attractive to read, but very important and crucial points are not addressed.
17. Were there other factors which could have influenced the presented results?
18. The authors report in the discussion section:” Interestingly, the severity and timing of the hematological disorder diagnosis did not significantly impact the risk of neurological outcomes in this study.” How did the authors established this? Please provide the information to the reader.
19. Furthermore, this is an outcome that should be reported in the result section. Please do so.
Tables:
See earlier remarks
The English grammar and style are fine throughout the entire article, minimal revision is required.
Author Response
Response to Reviewer Comments:
We would like to express our gratitude to the reviewer for their insightful feedback and constructive comments. We have carefully considered each point and provided responses and clarifications below.
Comment Number |
Reviewer's Comment |
Response |
1 |
Title Clarity: Suggest adding "cross-sectional study" to title |
It was proposed the design is mixed cohort " as per first reviewer suggestion " we added to the title |
2 |
Alignment of Aims: Discrepancy between aims and methods |
We revised the aim and added in the end of the introduction |
3 |
Inclusion Criteria Specificity: Specify when diagnoses occurred |
We provide additional details on when diagnoses were made to enhance clarity. |
4 |
Identification of Other Blood Disorders |
We provide a comprehensive list and details of other blood disorders. |
5 |
Grouping of Different Disorders |
As there is different haematological disorders as we aim to analyse consequences of haematological disorders and neurological outcome , despite different diagnosis , but all the disorders effects the blood flow and blood supply to the fetus |
6 |
Data Collection and Search Strategy |
The second researcher has access to the health institution’s database. It examined 4,365 health files over a period of 6 months, and 200 pregnant women with blood diseases were identified. These women were followed up and the children were evaluated neurologically at the age of 3 months after birth |
7 |
Assistant Details: Include number, training, robustness |
The second author who have access, also the confirmation was done by the second author |
8 |
Search Timing |
January 2022 to October 2022 |
9 |
Resolution of Anemia: Address management of resolved cases |
Mothers whose symptoms of the disease persisted throughout pregnancy, even with treatment, were selected. Only mothers who gave birth well despite suffering from a blood disease |
10 |
Neurological Outcomes: Provide details on evaluation |
We provide more comprehensive details on the evaluation of newborns for neurological outcomes. All new born were evaluated on the age of 3 months after delivery |
Results: |
||
11 |
Flow Diagram: Suggest inclusion for recruitment process |
We include a flow diagram to depict the recruitment process. |
12 |
Number of Included Participants: Ensure clear reporting |
the number of included participants stated n =200. |
13 |
Baseline Characteristics: Provide and analyze |
Data was not found for this |
14 |
Hematological Disorders Details: Offer more information |
All women received treatment , unless their conditions was chronic and persist across the pregnancy period |
15 |
Timing of Diagnosis: Investigate potential effects |
As data were identified from the hospital data bas we couldn’t assess the effect of time of diagnosis on the neurological outcomes, but we will consider testing this in separate study |
16 |
Definitions and Methodology: Enhance clarity |
We tried to enhance the clarity of definitions and provide more detailed explanations of our methodology. |
Discussion: |
||
17 |
Other Influencing Factors: Explore and discuss |
We will explore and discuss other potential factors that could have influenced our results. |
18 |
Severity and Timing Impact: Provide information |
Our suggestion of the effect of time should be modified , and will give us chance to be assess in in our next article We deleted this paragraph |
19 |
Outcome Reporting: Ensure accurate reporting in results |
We will ensure that outcomes, particularly those related to severity and timing of diagnosis, are accurately reported. We added paragraph about importance of early detection |
Round 2
Reviewer 1 Report
The work has been completely revised in many places in accordance with the reviewer's comments. The authors explained the research methodology in an understandable way. They currently describe it (line 110) as a retrospective-prospective cohort study “to assess the prevalence of neurological outcomes in newborns born to mothers with hematological examining the frequency and associations of health conditions within a defined population at a single moment, aligning with our research objectives.”
Data study design source, sample and collection have been changed, corrected according to the reviewer's comments and described perfectly. The inclusion of chart flow greatly enriched the paper and made it clear (line 109-181) The number of materials clearly defined, which is 200. Medical records that were found and included in the study were listed. All analysed data were described and discussed. The age at which the children development was assessed was determined (3 months – line 163). The results presented in Table 1 were briefly and correctly described in the text. The content contained in the lines 282-297 was presented in more detail than previously. It means that the potential mechanisms of hematological diseases that may affect the future development of children born to mothers with hematological diseases during pregnancy have been better explained. The last fragment of the conclusions was deleted and, in a modified form, included in the discussion, as I suggested.
Reviewer 2 Report
The authors have addressed the points and suggestions and significantly improved the quality of the article. The current meets the scientific quality required.
The authors should be complimented with these achievement.